# Creation and Validation of Patient-Derived Cancer Model Using Peritoneal and Pleural Effusion in Patients with Advanced Ovarian Cancer: An Early Experience

**DOI:** 10.3390/jcm13092718

**Published:** 2024-05-06

**Authors:** Ruri Nishie, Tomohito Tanaka, Kensuke Hirosuna, Shunsuke Miyamoto, Hikaru Murakami, Hiromitsu Tsuchihashi, Akihiko Toji, Shoko Ueda, Natsuko Morita, Sousuke Hashida, Atsushi Daimon, Shinichi Terada, Hiroshi Maruoka, Hiromi Konishi, Yuhei Kogata, Kohei Taniguchi, Kazumasa Komura, Masahide Ohmichi

**Affiliations:** 1Department of Obstetrics and Gynecology, Educational Foundation of Osaka Medical and Pharmaceutical University, 2-7 Daigakumachi, Takatsuki 569-8686, Osaka, Japan; ruri.nishie@ompu.ac.jp (R.N.); shunsuke.miyamoto@ompu.ac.jp (S.M.); hikaru.murakami@ompu.ac.jp (H.M.); hiromitsu.tsuchihashi@ompu.ac.jp (H.T.); akihiko.touji@ompu.ac.jp (A.T.); shouko.ueda@ompu.ac.jp (S.U.); natsuko.morita@ompu.ac.jp (N.M.); sosuke.hashida@ompu.ac.jp (S.H.); atsushi.daimon@ompu.ac.jp (A.D.); shinichi.terada@ompu.ac.jp (S.T.); hiroshi.maruoka@ompu.ac.jp (H.M.); hiromi.konishi@ompu.ac.jp (H.K.); yuhei.kogata@ompu.ac.jp (Y.K.); m-ohmichi@ompu.ac.jp (M.O.); 2Center for Medical Research & Development, Division of Translational Research, Osaka Medical and Pharmaceutical University, 2-7 Daigakumachi, Takatsuki 569-8686, Osaka, Japan; kohei.taniguchi@ompu.ac.jp (K.T.); kazumasa.komura@ompu.ac.jp (K.K.); 3Department of Regenerative Science, Okayama University Graduate School of Medicine, Dentistry and Pharmaceutical Sciences, 2-5-1 Shikata-cho, Kitaku, Okayama 700-8558, Okayama, Japan; pmes391t@s.okayama-u.ac.jp

**Keywords:** ovarian cancer, patient-derived xenograft, ascites, pleural effusion, sequence analysis

## Abstract

**Background**: The application of personalized cancer treatment based on genetic information and surgical samples has begun in the field of cancer medicine. However, a biopsy may be painful for patients with advanced diseases that do not qualify for surgical resection. Patient-derived xenografts (PDXs) are cancer models in which patient samples are transplanted into immunodeficient mice. PDXs are expected to be useful for personalized medicine. The aim of this study was to establish a PDX from body fluid (PDX-BF), such as peritoneal and pleural effusion samples, to provide personalized medicine without surgery. **Methods**: PDXs-BF were created from patients with ovarian cancer who had positive cytology findings based on peritoneal and pleural effusion samples. PDXs were also prepared from each primary tumor. The pathological findings based on immunohistochemistry were compared between the primary tumor, PDX, and PDX-BF. Further, genomic profiles and gene expression were evaluated using DNA and RNA sequencing to compare primary tumors, PDXs, and PDX-BF. **Results**: Among the 15 patients, PDX-BF was established for 8 patients (5 high-grade serous carcinoma, 1 carcinosarcoma, 1 low-grade serous carcinoma, and 1 clear cell carcinoma); the success rate was 53%. Histologically, PDXs-BF have features similar to those of primary tumors and PDXs. In particular, PDXs-BF had similar gene mutations and expression patterns to primary tumors and PDXs. **Conclusions**: PDX-BF reproduced primary tumors in terms of pathological features and genomic profiles, including gene mutation and expression. Thus, PDX-BF may be a potential alternative to surgical resection for patients with advanced disease.

## 1. Introduction

In recent years, the application of personalized treatments based on genetic information has been initiated in the field of cancer medicine [1]. Generally, genetic information is obtained from surgically resected cancerous tissues [2,3].

Patients with advanced cancer with peritoneal or pleural effusion have a poor prognosis [4]. In ovarian cancer (OC) and peritoneal cancer (PC), the initial treatment is debulking surgery. If it is impossible, interval debulking surgery after neoadjuvant chemotherapy may be performed [4]. In these situations, pathological diagnosis should be made before chemotherapy. Although these patients require early treatment, surgery may be difficult to perform because of poor activities of daily living (ADL) due to abdominal distention, a sense of fatigue, and poor general condition due to dehydration [5]. In many patients with OC and PC, early-stage detection is difficult to achieve. As a result, most patients have advanced disease at diagnosis [6]. According to the American Cancer Society, an estimated 12,810 women will die of the disease by 2022 [7]. The 5-year survival rate is 70% for International Federation of Gynecologists and Obstetricians (FIGO) stages I and II, and the rate decreases to 30% for stages III and IV [8]. Drainage of peritoneal or pleural effusion is often performed to relieve symptoms in patients with advanced OC and PC [9]. Notably, this treatment is minimally invasive compared with surgery [10].

Patient-Derived Xenograft (PDX) is a patient-derived cancer model prepared by transplanting a primary tumor from a patient into immunodeficient mice [11,12,13,14]. This is expected to be useful for personalized medicine. PDX prepared from tumors is reported to be pathologically and genetically similar to primary tumors in the colon [15], stomach [13], lung [16], head and neck [17], and uterus [11,12,18,19]. In OC, PDX has been reported to faithfully reproduce the primary tumor, including its clinical course [20,21,22,23]. Many studies have reported understanding the mechanisms of chemotherapy resistance, drug sensitivity tests, and precision medicine [24,25,26,27,28,29]. Studies related to PDX from body fluid (PDX-BF), including peritoneal and pleural effusion samples from pancreatic cancer [30], biliary tract cancer (BTC) [31], lung cancer [32], which is difficult to detect at the early stage and is lethal, have been published to develop drugs and validate biomarkers predictive of personalized therapy. According to these studies, PDX-BF can retain the histological characteristics of the tumor. Peritoneal and pleural effusion samples may also be useful as organoids and spheroids [4,33]. PDX-BF has been established for gynecological cancers, such as endometrial [34] and OCs [35,36]. In prior studies, similar pathological features, protein expression, and drug sensitivity were observed between primary tumors and PDX-BF; however, the genetic profile was not evaluated, and compared to PDX, there are fewer reports on refractory OC. PDX-BF may provide treatment based on genetic information to patients with advanced disease without surgery. The goals of the current study were to establish PDX-BF for patients with advanced OC and compare the histological and genetic similarities between PDX-BF and primary tumors.

## 2. Materials and Methods

### 2.1. Patient Information

This study included 23 patients with peritoneal or pleural effusion who underwent laparoscopic or open surgery at the Osaka Medical and Pharmaceutical University in Japan between March 2021 and May 2022. The patients underwent debulking surgery if considered possible. Other patients underwent laparoscopic salpingo-oophorectomy or biopsy for diagnosis. Eight patients were excluded due to negative cytology results for the peritoneal effusion sample. As a result, 15 patients were included in the study. None of the patients had undergone chemotherapy or radiation therapy before surgery. All patients were Asians with OC or PC. Written informed consent was obtained from all participants.

### 2.2. Tissue Samples

Approximately 800 mL of pleural effusion sample was collected from patients with respiratory distress and poor oxygenation. Approximately 500–3000 mL of peritoneal effusion sample was collected at surgery. During surgery, cancer tissues were obtained for PDX and further examination, as previously described [19]. Briefly, the primary tumor was collected, washed with saline solution, and divided into 500 mm^3^, 125 mm^3^, and 500 mm^3^ sections at the earliest time after surgical resection. The first section was placed in RNA later tissue storage reagent (Thermo Fisher Scientific, Waltham, MA, USA) for genetic analysis, stored overnight, and frozen at −80 °C. The second section was used to prepare the PDX, and the third was placed in 10% formalin for pathological analysis. In this study, PDX (F0) is defined as PDX derived from the cancer tissues, and PDX-BF (A0 or PE0) is defined as PDX derived from peritoneal and pleural effusion.

### 2.3. Establishment of Tumor-PDX (F0)

The tumor tissue was mixed with 2 mL of DMEM nutrient mix F-12 (DMEM/F12, Gibco, Thermo Fisher Scientific) and Matrigel (Corning, New York, NY, USA) and cut into small pieces with scissors. The fragmented tumor was injected subcutaneously into the backs of immunocompromised mice using a 22-gauge needle (Figure 1).

### 2.4. Establishment of PDX-BF (PE0/A0)

Peritoneal and pleural effusion samples were divided into 50 mL sterile tubes immediately after collection. Thereafter, 100 units per 20–30 mL of heparin were added to the tubes. The samples were centrifuged at 3000 min^−1^ for 3 min, and the supernatants were removed. This procedure was performed several times to ensure the collection of cancer cells only. The isolated cells were mixed with 2 mL of DMEM/F12 and Matrigel and injected subcutaneously into the backs of immunocompromised mice (Figure 1).

### 2.5. Animals

All mice used in this study were obtained with permission from the Ethics Committee of Osaka Medical and Pharmaceutical University (Assurance Number 21007-A). Female NOD.CB17-PrkdcSCID/J mice (aged 4–8 weeks old; Oriental BioService, Kyoto, Japan) were used. These animals were housed in a specific pathogen-free barrier facility at 24–26 °C with a humidity of 30–50% and free access to sterile water and standard rodent chow. If animal killing was required, trained staff performed the cervical dislocation and killing.

### 2.6. Sampling of the Xenograft Tumor

Mice with engrafted tumors were killed and the sampled tumors were divided into 125 mm^3^, 500 mm^3^, and 125 mm^3^ sections. The first section was placed in RNA later for genetic analysis. The second section was placed in 10% formalin and embedded in blocks within 72 h for paraffin sectioning. The third section was placed in a stem cell banker, frozen in liquid nitrogen, and stored at −80 °C for long-term preservation.

### 2.7. Pathological Analysis Using Immunohistochemistry

Paraffin sections of primary and xenograft tumors were subjected to H&E staining and immunostaining. Immunostaining was performed using the enzyme antibody method. The primary antibodies used to evaluate the epithelial area were AE1/AE3 (67306, 1:50 dilution, Cell Signaling Technology, Danvers, MA, USA), p53 (2527, 1:100 dilution; Cell Signaling Technology) due to its high expression in OC, and Ki-67 (9027, 1:200 dilution; Cell Signaling Technology) to assess cell proliferative potential. CD10 (65534, 1:100 dilution, Cell Signaling Technology) was used to evaluate the stromal cells. Each secondary antibody was specific to the primary antibody. Pathological images of the tissues were obtained using a microscope (BZ-X700 Series, Keyence, Osaka, Japan). WinROOF 2021 was used to determine the Ki-67 positivity rate.

### 2.8. Sample Preparation for DNA and RNA Sequencing

The MagMAX DNA Multi-Sample Ultra 2.0 Kit and MagMAX mirVana Total RNA Isolation Kit (Thermo Fisher Scientific) were used to extract DNA and RNA from patients and PDX tumors stored in RNA. The extracted DNA and RNA were quantified using a Qubit4 Fluorometer (Thermo Fisher Scientific), Qubit™ 1×dsDNA HS Assay Kits, and Qubit™ RNA HS Assay Kits (Thermo Fisher Scientific). RNA was reverse-transcribed to cDNA using a Gene Amp PCR System 9700 (Thermo Fisher Scientific) and SuperScript™ IV VILO™ Master Mix (Thermo Fisher Scientific).

### 2.9. Library Prepare

Libraries were prepared using an automated Ion Chef System (Thermo Fisher Scientific). The Ion AmpliSeq Cancer Hotspot Panel v2 (Thermo Fisher Scientific) was used to detect the genetic mutations. This panel was designed to amplify 207 amplicons covering approximately 2800 COSMIC mutations from 50 oncogenes and tumor suppressor genes. The Ion AmpliSeq RNA Cancer Panel (Thermo Fisher Scientific) was used to analyze gene expression. This panel was designed to amplify 50 amplicons targeting transcript sequences from 50 oncogenes and tumor suppressor genes. The libraries were prepared using these panels and an Ion AmpliSeq Kit for Chef DL8 (Thermo Fisher Scientific). Library concentrations were measured using QuantStudio 5 (Thermo Fisher Scientific) and the Ion Library TaqMan Quantitation Kit (Thermo Fisher Scientific). All concentrations were adjusted to 50 pmol/L.

### 2.10. Run for DNA and RNA Sequencing

Emulsion PCR and chip loading were performed on the Ion Chef System using Ion 540 Kit-Chef. Sequencing analysis was performed using Ion GeneStudio S5 Prime and the kit mentioned above.

### 2.11. Data Analyses

Sequence data were analyzed using Torrent Suite Software v5.18.0 and Ion Reporter v5.18 software. Python (3.9.13) was used as the programming language. DNA was detected in VCF files with a variant allele frequency of ≥10% and annotated using ANNOVAR (20210202). The RNA gene expression levels were normalized to reads per million (RPM). The figures were produced using paplot (0.5.5). SHIROKANE, provided by the Human Genome Center, Institute of Medical Science, University of Tokyo, was the supercomputer used in this study.

### 2.12. Statistical Analyses

JMP Pro v15 (SAS Institute Japan, Tokyo, Japan) was used for statistical analyses. Continuous variables were compared using the Mann–Whitney U-test and expressed as medians (interquartile range). Categorical variables were compared using Fisher’s exact test. Pearson’s correlation coefficient was calculated to determine the degree of association between primary tumors and xenografts. Statistical significance was set at *p* < 0.05.

## 3. Results

### 3.1. Establishment of PDX Tumor

PDX and PDX-BF were created following the method shown in Figure 1. Table 1 shows details of the 15 cases and a comparison of the factors affecting the success rate of xenograft establishment. All patients were diagnosed with OC and PC. In terms of FIGO stage, there were 7 cases with stage III disease and 8 cases with stage IV disease. PDX-BF was established for 8 of the 15 cases, with a success rate of 53%. The median latency period was 5 (2–7) weeks. PDX was established from primary tumors in 6 of the 13 cases, with a success rate of 46%. The latency period was 10 (4–16) weeks. For the engrafted and failed PDXs, age, HRD/BRCA status, CA125 level, follow-up period, and Progression-Free Survival (PFS) were not found to differ significantly. Of the eight cases of engraftment tumors from peritoneal or pleural effusion samples, seven were discussed in this study.

The clinical characteristics of the 7 patients are shown in Table 2. Pathologically, there were four cases of high-grade serous carcinoma, one carcinosarcoma, one low-grade serous carcinoma, and one clear cell carcinoma. All patients had FIGO stage III or IV disease. Cases 1, 2, 5, 6, and 7 underwent debulking surgery, while cases 3 and 4 underwent laparoscopic biopsy. The median age of patients was 64 (63–66). Genetic testing was performed in six cases. Among them, two cases (i.e., a high-grade serous carcinoma case and a carcinosarcoma case) were positive for HRD (wild-type BRCA).

### 3.2. Histological Evaluation

Figure 2 shows the histological findings for the primary tumors (P), PDX (F0), and PDX-BF (A0 or PE0). In two cases of high-grade serous carcinoma, similar features were found based on H&E and immunohistochemical staining. In fact, papillary growth and a fibrous branch were observed. The nuclei showed strong atypia. Immunohistochemistry revealed similar results. In particular, the cancer cells were found to be positive for AE1/AE3 and negative for p53. The Ki-67 labeling of P0, F0, and A0 was 20%, 31%, and 27% in case 1 and 21%, 33%, and 21% in case 3, respectively (Figure 2a,b). Figure 2c shows details regarding case 5 (carcinosarcoma). The primary tumor had two malignant components: epithelial and sarcomatous components. The high-grade epithelial carcinoma component was positive for AE1/AE3, and the sarcomatous component was positive for CD10. Interestingly, PDX-BF displayed pathological and immunohistochemical features similar to those of the primary tumor. P53 was negative in both tumors. The Ki-67 labeling was 26% in P and 27% in A0. Figure 2d shows details regarding case 6 (low-grade serous carcinoma). Similar pathological and immunohistochemical features were found between P and A0 tumors; the tumor cells were small papillae-containing cells with uniform nuclei and inconspicuous mitotic activity. The tumor cells were positive for AE1/AE3 and negative for p53. Ki-67 labeling in P and A0 cells was 6%. Figure 2e shows details regarding case 7 (clear cell carcinoma). Similar pathological and immunohistochemical features were observed among P, F0, and PE0. The tumors exhibited a solid pattern. The solid architecture was composed of sheets of clear cells separated by delicate septa. The tumor cells were positive for AE1/AE3 and negative for p53. Ki-67 labeling was 7% in P, 7% in F0, and 6% in PE0.

### 3.3. Gene Mutation Analysis

Figure 3a shows a heat map of the genetic mutations in the seven cases. All cases had genetic mutations in tumor protein p53 (TP53), platelet-derived growth factor alpha (PDGFRA), erb-b2 receptor tyrosine kinase 4 (ERBB4), and adenomatous polyposis coli (APC). Six common mutations were found among P, F0, and A0 in cases 1 and 2, while four common mutations were found among P, F0, and A0 in cases 3 and 4. Five common mutations were found between P and A0 in case 5, while six were found in case 6. Six common mutations were found among P, F0, and PE0 in case 7. In all cases, most genetic mutations in the primary tumors were present in xenografts. The number of genetic mutations tended to be higher in xenografts than in primary tumors. Mutations in the serine/threonine kinase 11 gene (STK11), NRAS, smoothened (SMO), and phosphatase and tensin homolog (PTEN) were present in almost all xenografts.

The correlations between variant allele frequencies (VAFs) for cases 1, 2, 3, and 4 are shown in Figure 3b. The Pearson’s correlation coefficients for P and F0 were 0.987, 0.939, 0.565, and 0.996, respectively. The correlation coefficients of P and A0 were 0.746, 0.935, 0.564, 0.961, while those of F0 and A0 were 0.673, 0.764, 0.996, and 0.961, respectively. A strong positive correlation was found between P and A0 in cases 1, 2, and 4. The correlations of the VAFs in cases 5 and 6 are shown in Figure 3c. The Pearson’s correlation coefficients of P and A0 were 0.901 and 0.998, respectively, indicating a strong positive correlation in both cases. Figure 3d shows the VAFs of case 7 using pleural effusion sample. The correlation coefficient of P and F0 was 0.518, that of P and PE0 was 0.578, and that of F0 and PE0 was 0.948. A strong positive correlation was found between F0 and PE0.

### 3.4. Gene Expression Analysis

A heat map of gene expression for the seven cases is shown in Figure 4a. High-expression genes are shown in red, and low-expression genes are shown in blue. In all cases, guanine nucleotide-binding protein, alpha-stimulating activity polypeptide 1 (GNAS), and nucleophosmin 1 (NPM1) were highly expressed, while fms-related tyrosine kinase 3 (FLT3) and hepatic nuclear factor 1α (HNF1A) exhibited low expression. In cases 1, 2, and 3, the gene expression levels were similar among P, F0, and A0. In cases 5 and 6, the gene expression levels were similar between P and A0. In cases 4 and 7, the gene expression levels were similar between P and F0 but different between A0 and PE0, especially the expression of ERBB4, fibroblast growth factor receptor 2 (FGFR2), cadherin 1 (CDH1), erb-b2 receptor tyrosine kinase 2 (ERBB2), fibroblast growth factor receptor 3 (FGFR3), MET, epidermal growth factor receptor (EGFR), and SMO.

The correlation between gene expression levels is outlined in Figure 4b–d. All results indicated positive correlations. The correlation coefficients for cases 1, 2, 3, and 4 are shown in Figure 4b. The values for P and F0 were 0.897, 0.830, 0.773, and 0.851, respectively. The values for P and A0 were 0.820, 0.820, 0.719, and 0.620, respectively. The coefficients between P and A0 revealed a strong positive correlation in cases 1, 2, and 3 and almost similar correlation coefficients compared to those of F0. Figure 4c shows the correlation in gene expression between P and A0 in cases 5 and 6. The correlation coefficients between P and A0 were 0.760 and 0.783 in cases 5 and 6, respectively, indicating a strong correlation. The correlation in gene expression in case 7 is shown in Figure 4d. The correlation coefficient was 0.417 between P and F0 and 0.644 between P and PE0. That between F0 and PE0 was 0.508.

## 4. Discussion

In the current study, PDX-BF was established, with a success rate of 53% for OC and PC. Similar pathological and immunohistochemical features were observed between primary tumors and PDX-BF. DNA and RNA sequencing revealed similar profiles of genetic mutations and expression between primary tumors and PDX-BF.

The orthotopic models resemble original human cancers concerning histology, vasculature, gene expression, response to chemotherapy, and metastatic biology. In comparison to subcutaneous models, orthotopic models generally show the appropriate metastatic pattern associated with each disease. Orthotopic models play an essential role in cancer research associated with tumor growth, invasion, metastasis, and microenvironment because they have a similar tumor microenvironment to the primary tumors [37,38]. However, the aim of the current study is to create a relatively large tumor from body fluid, mainly from ascites, and compare the pathological features and genetic profiles to the primary tumor. Then, we chose the subcutaneous implantation because of its simplicity and ease of confirming tumor implantation. Subcutaneous implantation also allowed us to observe the growth rate of the tumors and to compare the differences in tumor growth between PDX and PDX-BF.

PDX-BF has been established from several organs, including the ovary [35,36], uterus [34], pancreas [30,39], kidney [40], lung [32], biliary tract [31], and stomach [13]. The engraftment rate is important for preparing xenografts from peritoneal or pleural effusion samples. The PDX engraftment rate in OC has been reported to be 25–90% [21,22,23,24,25,26,41]. The latency periods for tumor growth were 1–12 months [21,22,24,41]. Weroha et al. reported that the better growth rate was due to advanced stage, high-grade tumors, and the presence of ascites [41]. PDX-BF was established for 29 and 33 cases of OC [35,36], 1 case of endometrial cancer [34], 3 and 12 cases of pancreatic cancer [30,39], 1 case of renal cancer [40], 2 cases of lung cancer [32], 3 cases of BTC [31], and 4 cases of gastric cancer [13]. The success rates were 29% and 31% for OC [35,36], 70% for pancreatic cancer [30], 60% for BTC [31], and 24% for gastric cancer [13]. The latency periods for tumor growth were 2–12 months and 2–3 months in OC [35,36], 2 months in endometrial cancer [34], and 2–12 weeks in pancreatic cancer [30]. For the factors contributing to the growth rate, these studies revealed successful engraftment for chemotherapy-resistant and poor prognosis cases [13,31,34,39,40]. Few reports have compared PDX and PDX-BF in the same cases. Kang et al. reported that BTC PDX-BF may have a higher engraftment rate than PDX (60% vs. 5.8%) [31,42,43]. Although the cause is unknown, this higher rate may be due to differences in the microenvironment between tumors and peritoneal and pleural effusion [31]. In the current study, the success rate and latency period were 53% and 5 weeks in PDX-BF and 46% and 10 weeks in PDX, respectively; PDX-BF had a higher success rate and shorter latency periods than PDX. Peritoneal and pleural effusion samples are predicted to contain fewer impurities than tumors, contributing to a high engraftment rate. That could be considered potential differences between PDX-BF and PDX.

PDXs have been reported to retain the histological features of the primary tumor [13,15,16]. It is also approximated in OC [21,22,23,24,25]. According to several authors, PDX-BF maintains the histological features of the primary tumor. In this study, the PDX had histological characteristics of the primary tumor in all cases. Interestingly, PDXs-BF from patients with carcinosarcoma displayed pathological features similar to those of the primary tumor. Two components were identified, namely epithelial and sarcomatous components. PDX has been prepared from carcinosarcoma tumors in several prior studies. Glaser et al. found that PDX prepared from tumors mainly contained epithelial elements and did not retain many sarcoma components. This result is because the sarcoma component of xenograft tissue was derived from the mouse host, and the rapid growth of PDX tumors may not permit concurrent outgrowth of the sarcoma components [44]. We established PDX and PDX-BF for patients with clear cell carcinoma. Studies have also been published on PDX from clear cell carcinoma in cervical cancer and renal cancer [45,46]. According to Serebrenik et al., PDX from ovarian clear cell carcinoma had histologically similar characteristics to primary tumors [47]. In the current study, PDX-BF prepared from pleural effusion sample had the same structural features as clear cell carcinoma (i.e., the primary tumor). The results suggest that diagnosis using peritoneal and pleural effusion samples might be available for advanced OC.

There have been several reports of genetically investigating PDX in OC. Cybulska et al. reported that genetic mutations that are known to be frequent in OC were found in PDX as well [48]. Dong et al. demonstrated that the two-passage PDX tumors were differently expressed in 130 genes from the primary tumors. It was considered altered to adapt to the mouse host [21]. Some reports indicate that the clinical course of drugs is consistent with the primary tumors and PDXs [48]. And Topp et al. reported that the expression of CCNE1, LIN28B, and BCL-2 genes correlate with drug resistance [26]. In addition, there are some reports on drug sensitivity tests [24,25,26,27,28,29,49,50]. Odunsi et al. reported that the combination of immune checkpoint inhibitors, considered to have limited efficacy in treating OC, and adoptive cell transfer (ACT) therapy prevented the growth of PDX tumors [51]. And there is a report that using paclitaxel plus itraconazol in combination with PDX of carcinosarcoma, which is rare, inhibited tumor growth [52]. Thus, PDX is expected to lead to novel therapeutic developments.

Some studies have reported gene mutation analyses of PDX-BF. These models retained fidelity to the patient’s tumor. Liu et al. performed a whole-exome sequencing analysis of primary tumors and PDX-BF in high-grade serous OC. A total of 82 mutations were found in the primary tumor, and all mutations were present in PDX-BF [35]. Golan et al. performed whole-genome sequencing of pancreatic cancers. Mutations were similar between primary tumors and PDX-BF [30]. Lee et al. performed gene expression profiling in renal cancer and demonstrated high concordance between PDX-BF and PDX in renal cancers [40]. By performing whole-exome sequencing in non-small-cell lung cancer, Xu et al. revealed that the xenograft and primary tumor genotypes were similar [32]. Kang et al. successfully established patient-derived cancer cell cultures and xenograft models using malignant peritoneal effusion samples from patients with BTC. DNA sequencing was performed, and 51 genetic alterations were identified in 48 genes from the primary tumor, patient-derived cancer cell cultures, and PDX models. These researchers concluded that the pattern of commonly mutated genes in these models differed from that in commercially available BTC cell lines [31]. In the current study, TP53, PDGFRA, ERBB4, and APC were detected in all primary tumors and xenografts. In contrast, PDXs and PDXs-BF tended to have more gene mutations than the primary tumors. The PDX and PDX-BF models tended to have mutations in STK11, NRAS, SMO, and PTEN, whereas the primary tumors did not have such mutations. STK11 is a cancer suppressor gene that has been demonstrated to be associated with poor prognosis [53,54]. RAS is an oncogene that occurs in approximately 19% of cancers [55]. SMO is an oncogene that has been reported to be an independent prognostic factor that can predict better clinical outcomes in patients receiving immune checkpoint therapy [56]. PTEN is a cancer suppressor gene whose importance in tumorigenesis is underscored by its frequent mutations in human cancer [57]. Induction of these mutations may be an important characteristic of PDX and PDX-BF. These differences between original tumors and PDXs may occur in the reaction of tumor cells due to the influence of the mouse tissue. Otherwise, the replacement of stroma by mouse-derived tissue may have affected the results.

Although several problems still need to be resolved, including low success rates and long implantation and drug testing periods, the results of the current study showed that PDX-BF could be an alternative to surgical specimens. Thus, genetic-based medicine may be provided to patients with difficulties with surgery. The results of this study suggest additional possibilities for future development. Notably, peritoneal and pleural effusion samples can be collected several times. Moreover, the procedure is easier and less painful than a biopsy [58,59,60].

The current study had several limitations. First, the sample size was small, and demonstrating statistical significance as a factor in the preparation of PDX-BF proved difficult. Second, DNA and RNA gene analyses were performed via targeted sequencing instead of whole-genome sequencing. As a result, only a few genes were detected. Third, we did not compare PDX-BF and cell line-derived xenograft. Fourth, although several gene mutations that could be a target for therapy were identified, a drug efficacy test was not performed. Based on these problems, studies, including cell line-derived xenograft and drug sensitivity testing, should be performed to establish personalized medicine.

## 5. Conclusions

Patient-derived cancer models prepared from peritoneal and pleural effusion in patients with advanced malignancies recapitulate the primary tumor in some respects. Especially in patients with OC and PC, peritoneal and pleural effusion might be an alternative for surgically removed tumor tissues.

## Figures and Tables

**Figure 1 jcm-13-02718-f001:**
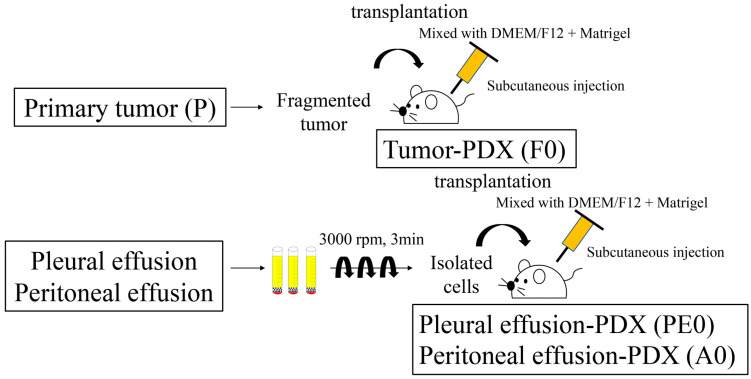
Protocols used to prepare Patient-Derived Xenografts (PDXs) using the primary tumor and peritoneal and pleural effusion samples. The tissue of primary tumors was immediately washed and minced. The tumor fragments mixed with DMEM nutrient mix F-12 (DMEM/F12) and Matrigel were subcutaneously injected into immunodeficient mice (F0). The collected peritoneal and pleural effusion samples were centrifuged several times at 3000 min^−1^ for 3 min. Isolated cancer cells with DMEM/F12 and Matrigel were subcutaneously injected into immunodeficient mice (PE0 or A0).

**Figure 2 jcm-13-02718-f002:**
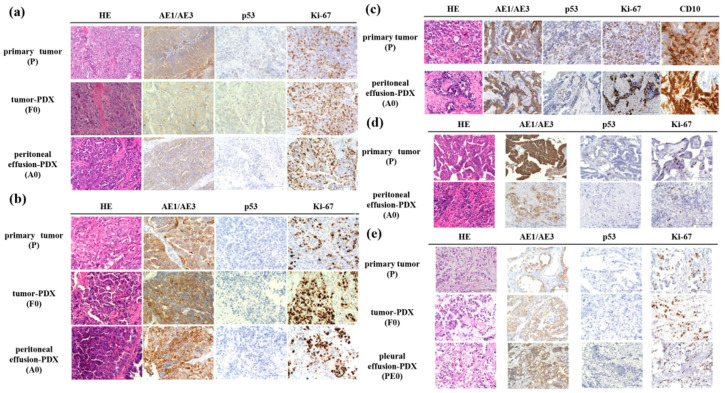
Pathological and immunohistochemical findings in primary tumors (P), patient-derived xenografts (PDXs, F0), and PDX from pleural effusion (PE0) and peritoneal effusion (A0) samples. (**a**,**b**) Similar pathological findings among P, F0, and A0 for cases 1 and 3 (i.e., serous carcinoma). The tumor cells mainly had a solid architecture. Papillary growths and a fibrous branch were often observed. The nuclei had strong atypia. The tumor cells were positive for AE1/AE3 and negative for p53. The percent Ki-67 labeling of each tumor ranged from 30% to 33%. (**c**) In case 5 (i.e., carcinosarcoma), the primary tumor had two malignant components: epithelial and sarcomatous components. The high-grade epithelial carcinoma component was positive for AE1/AE3, while the sarcomatous component was positive for CD10. Interestingly, similar pathological and immunohistochemical features were found between P and A0. The percent Ki-67 labeling was 26% in P and 27% in A0. (**d**) In case 6 (i.e., low-grade serous carcinoma), similar pathological and immunohistochemical features were found between P and A0; the tumor cells were small papillae-containing cells with uniform nuclei and inconspicuous mitotic activity. The tumor cells were positive for AE1/AE3 and negative for p53 and had 6% Ki labeling. (**e**) In case 7 (i.e., clear cell carcinoma), similar pathological and immunohistochemical features were found among P, F0, and PE0. The tumors exhibited a solid pattern. The solid architecture consisted of sheets of clear cells separated by delicate septa. The tumor cells were positive for AE1/AE3 and negative for p53. Ki-67 labeling was 7% in P, 7% in F0, and 6% in PE0. The scale bar indicates 100 µm.

**Figure 3 jcm-13-02718-f003:**
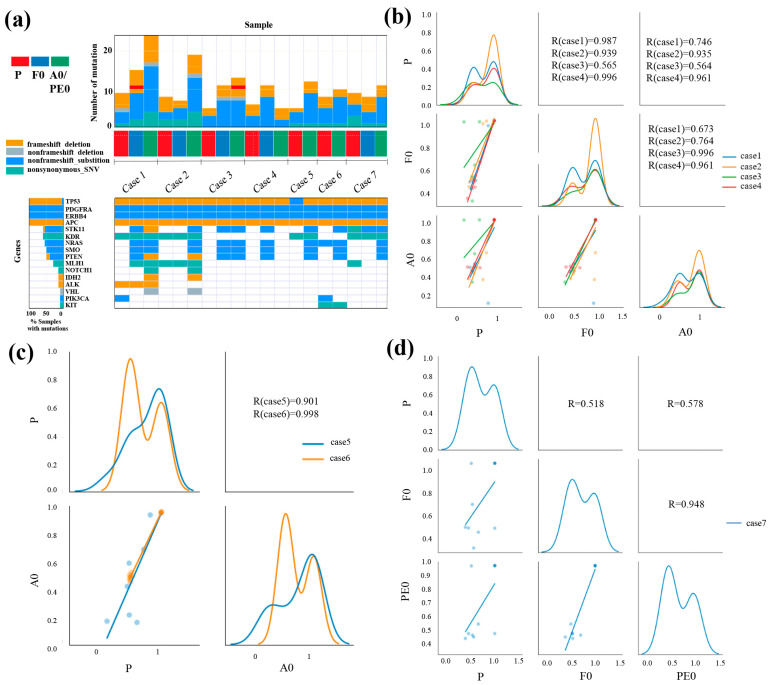
Summary of the relationships between somatic mutations in primary tumors and matched xenograft tumors based on the DNA-seq results. (**a**) Heat map analysis of DNA profiling in primary tumors (P), patient-derived xenografts (PDXs, F0), and PDXs from body fluid, including peritoneal effusion (A0) and pleural effusion (PE0) samples, using Ion AmpliSeq Cancer Hotspot Panel v2. In all cases, xenograft tumors had almost the same mutation as the primary tumors. (**b**–**d**) Variant allele frequencies (VAFs) of somatic mutations identified in primary and xenograft tumors. Scatter plots and linear regression of the VAFs levels in the primary and xenograft tumors. This diagonal graph shows the kernel density estimation (KDE). In all cases, primary and xenograft tumors show a positive correlation.

**Figure 4 jcm-13-02718-f004:**
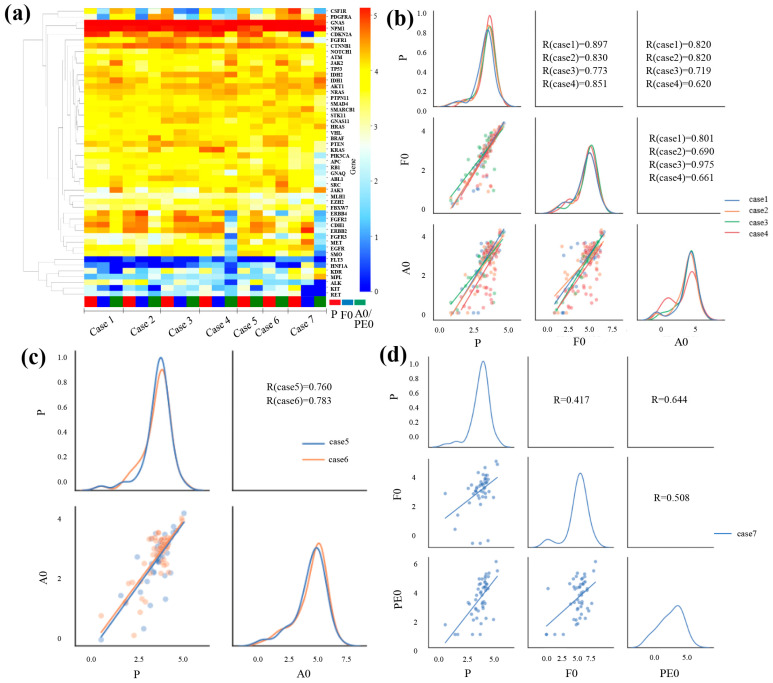
Gene expression in primary tumors (P), patient-derived xenograft (PDX, F0), and PDX from body fluid (A0 or PE0) based on RNA sequencing. (**a**) Clustering and heat map analysis of mRNA profiling in tissues of the primary and xenograft tumors using Ion AmpliSeq RNA Cancer Panel. In most cases, the gene expression in primary and xenograft tumors exhibited a similar pattern, whereas in cases 4 and 7, PDX-BF exhibited partially different results. (**b**–**d**) Pair plot showing gene expression in primary and xenograft tumors. Scatter plots and linear regression of the gene expression in the primary and xenograft tumors. The diagonal graph shows the kernel density estimation (KDE). In all cases, primary and xenograft tumors show a positive correlation. The normalized data have been changed to base 10 logarithms and z-score.

**Table 1 jcm-13-02718-t001:** Characteristics of the study participants.

		Overall	Established	Failed	*p* Value
Number of patients		15	8 (53%)	7 (47%)	
Age *, years old		64 (61–72)	64 (62–66)	71 (57–78)	0.3
Histological type	high-grade serous carcinoma	10	5	5	
	carcinosarcoma	1	1	0	
	low-grade serous carcinoma	1	1	0	
	clear cell carcinoma	3	1	2	
FIGO stage	III	7	3	4	
	IV	8	5	3	
HRD positive		6	3	3	1.0
BRCA positive		1	0	1	
CA125 *, U/mL		1138 (224–3117)	865 (288–3344)	1709 (115–3117)	0.9
Follow-up *, months		11 (10–18)	11 (8–12)	16 (11–20)	0.03
PFS *, months		10 (7–12)	10 (7–11)	11 (7–18)	0.4

* median (interquartile range). FIGO, International Federation of Gynecology and Obstetrics; HRD, Homologous Recombination Deficiency; PFS, Progression-Free Survival.

**Table 2 jcm-13-02718-t002:** Characteristics of the seven patients discussed in this study.

							PDX Growth		
Case	Age	Type of Cancer	Histological Type	FIGO	HRD	Source	Tumor (F0)	Ascites (A0)	Pleural Effusion (PE0)
1	65	PC	high-grade serous carcinoma	3C	positive (GIS 42)	ascites	Yes	Yes	-
2	63	PC	high-grade serous carcinoma	4B	negative	ascites	Yes	Yes	-
3	64	OC	high-grade serous carcinoma	4B	negative	ascites	Yes	Yes	-
4	66	PC	high-grade serous carcinoma	4A	negative	ascites	Yes	Yes	-
5	61	OC	carcinosarcoma	3C	positive (GIS 43)	ascites	No	Yes	-
6	72	OC	low-grade serous carcinoma	3B	negative	ascites	No	Yes	-
7	63	OC	clear cell carcinoma	3B	unknown	pleural effusion	Yes	-	Yes

PDX, Patient-derived xenograft; FIGO, International Federation of Gynecology and Obstetrics; HRD, Homologous Recombination Deficiency; PC, Peritoneal cancer; OC, Ovarian cancer; GIS, Genomic Instability Status.

## Data Availability

The data presented in this study are available upon request from the corresponding author.

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
