# Peer review of "Creation and Validation of Patient-Derived Cancer Model Using Peritoneal and Pleural Effusion in Patients with Advanced Ovarian Cancer: An Early Experience"

_jcm, 2024, doi:10.3390/jcm13092718_

Round 1

Reviewer 1 Report

Comments and Suggestions for Authors

Dear authors,

Thanks for your good work

A few remarks to be noted -from the clinical point of view-

1- Title: Please add the words "a pilot study" or "early experience"

2- Introduction:

- Some concepts need to be more clear. In ovarian cancer management, the opposite of surgery in advanced ovarian cancer surgery is neoadjuvant therapy for downstaging then surgery not simple paracentesis

- Pancreatic cancer with ascites or effusion is a different issue with very short survival and dismal prognosis and the main treatment is mainly palliative. Also, surgery in such a condition is never an option

3- Methods:

- Including patients who underwent surgery despite having positive ascites or pleural effusion is non-practical. Although the presence of positive ascites or effusion does not prevent surgery, it can not be achieved in all these categories of patients especially if heavy peritoneal and omental infiltration is present

- How can laparoscopic optimal surgery be done in such a patient?! As the authors mentioned they included patients with laparoscopic surgery

- What about the role of MDT in such patients? Why did these patients not receive neoadjuvant chemotherapy?

4- Results: The same issue again, how did the patients with stage IVb high-grade serous carcinoma undergo upfront surgery?

5- Discussion: There are two conclusions sections. One at the end of the discussion and one under the subtitle "conclusion"

Author Response

We appreciate the time and effort of the editor and referees in reviewing our manuscript. We have addressed all the issues indicated in the review report and hope that the revised version meets the journal’s requirements for publication.

Response to Comments from Reviewer 1:

Comment 1:

Title: Please add the words "a pilot study" or "early experience"

Response:

As you pointed out, we changed the title as follows.

Creation and validation of patient-derived cancer model using peritoneal and pleural effusion in patients with advanced ovarian cancer: A early experience

Comment 2:

Introduction:

Some concepts need to be more clear. In ovarian cancer management, the opposite of surgery in advanced ovarian cancer surgery is neoadjuvant therapy for downstaging then surgery not simple paracentesis.

Response:

The primary management of ovarian cancer is debulking surgery followed by chemotherapy. In cases in which debulking surgery is not feasible, exploratory laparotomy is performed for diagnostic purposes. In advanced ovarian cancer, it has not been established whether surgery or chemotherapy should proceed. In our hospital, we perform debulking surgery or exploratory laparotomy in operable patients. However, surgery is difficult for patients with peritoneal and pleural effusion. Suppose pathology can be diagnosed using peritoneal and pleural effusion, as was studied here. In that case, it may be possible to do chemotherapy as early as possible using peritoneal and pleural effusion instead of exploratory laparotomy. As you pointed out, we added the sentences about ovarian cancer management. (page2, lines 49-52)

Comment 3:

Introduction:

Pancreatic cancer with ascites or effusion is a different issue with very short survival and dismal prognosis and the main treatment is mainly palliative. Also, surgery in such a condition is never an option.

Response:

As you pointed out, in cases of pancreatic and other cancers except ovarian cancer, patients with ascites or effusion are often not operated on and are transferred to palliative, which is different from advanced ovarian cancer. The report on pancreatic cancer is presented as an example of PDX-BF. It produced PDX-BF to develop drugs and validate biomarkers predictive of personalized therapy. As the objectives are different, we added the sentences. (page2, lines 70-71)

Comment 4:

Methods:

Including patients who underwent surgery despite having positive ascites or pleural effusion is non-practical. Although the presence of positive ascites or effusion does not prevent surgery, it can not be achieved in all these categories of patients especially if heavy peritoneal and omental infiltration is present.

Response:

In advanced ovarian cancer, it has not been established whether surgery or chemotherapy should proceed. In our hospital, operable patients underwent debulking surgery or exploratory laparotomy, followed by chemotherapy. The present study examined cases assessed as operable and underwent laparoscopic or open surgery. As you pointed out, we added the sentences about it. (page2, lines 88-90)

Comment 5:

Methods:

How can laparoscopic optimal surgery be done in such a patient? As the authors mentioned they included patients with laparoscopic surgery

Response:

If debulking surgery was considered possible, surgery was done by laparotomy, and if not, laparoscopic biopsy was done. As you suggested, we added the sentences. (page2, lines 88-90)

Comment 6:

Methods:

What about the role of MDT in such patients? Why did these patients not receive neoadjuvant chemotherapy?

Response:

As you mentioned, some patients with ovarian cancer receive MDT at initial diagnosis. In the current study, all the participants received open debulking surgery or laparoscopic salpingo-oophorectomy or biopsy for diagnosis; the patients with MDT were excluded.

Comment 7:

Results

The same issue again, how did the patients with stage IVb high-grade serous carcinoma undergo upfront surgery?

Response:

The patient received exploratory laparotomy, and the surgeon diagnosed debulking surgery was not possible. She received a laparoscopic left adnexectomy. Based on the pathology results, she received chemotherapy. As you pointed out, we added sentences about the details of surgery. (page5, lines 214-215)

Comment 8:

Discussion:

There are two conclusions sections. One at the end of the discussion and one under the subtitle "conclusion."

Response:

As you suggested, we deleted the end of the discussion.

Reviewer 2 Report

Comments and Suggestions for Authors The authors used a patient xenograft method from peritoneal and pleural effusions from patients with ovarian cancer who had positive cytologic findings. The pathological findings based on immunohistochemistry were compared between primary tumor and patient-derived xenograft from body fluid. In addition, genomic profiles and gene expression were assessed using DNA and RNA sequencing to compare primary tumors, PDX and PDX-BF.
The article is well organized, conceptually well designed and may represent a step forward in personalized medicine in gynecologic pathology.
However, it is not clear from the methodology which criteria the authors used to select the antibodies for pathological analysis by immunohistochemistry. The authors used the AE1/AE3 antibody, which is a universal epithelial marker. A specific selection of markers for ovarian cancers analyzed in the article would be CK7, p53 and PAX8. I think the choice of immunohistochemical markers should be clarified.  

Author Response

We appreciate the time and effort of the editor and referees in reviewing our manuscript. We have addressed all the issues indicated in the review report and hope that the revised version meets the journal’s requirements for publication.

Response to Comments from Reviewer 2:

Comment 1:

The authors used a patient xenograft method from peritoneal and pleural effusions from patients with ovarian cancer who had positive cytologic findings. The pathological findings based on immunohistochemistry were compared between primary tumor and patient-derived xenograft from body fluid. In addition, genomic profiles and gene expression were assessed using DNA and RNA sequencing to compare primary tumors, PDX and PDX-BF.

The article is well organized, conceptually well designed and may represent a step forward in personalized medicine in gynecologic pathology.

However, it is not clear from the methodology which criteria the authors used to select the antibodies for pathological analysis by immunohistochemistry. The authors used the AE1/AE3 antibody, which is a universal epithelial marker. A specific selection of markers for ovarian cancers analyzed in the article would be CK7, p53 and PAX8. I think the choice of immunohistochemical markers should be clarified. 

Response:

AE1/AE3 was used to confirm that PDX is an epithelial cancer and to identify the epithelial part in a case of carcinosarcoma. The CK7 was not used in this study because AE1/AE3 are antibodies that react with various cytokeratins. The p53 was used for its high expression in ovarian cancer, and Ki-67 was used to assess cell proliferative potential.

As you pointed out, PAX8 is a specific selection of markers for ovarian cancers. It was not used because the purpose of this sutdy was to ensure that primary tumors, PDX (F0) and PDX-BF (A0 and PE0), were comparable.

We added the sentences about the choice of immunohistochemical markers. (page4, lines 150-151)